# The Origin of the Magnetic and Electric Dipole Moments of $Ni^{2+}$ in $NiCr_2O_4$

**Mikhail Eremin *** and **Kirill Vasin**

Institute of Physics, Kazan Federal University, Kazan 420008, Russia
* Correspondence: meremin@kpfu.ru

**Abstract:** The energy level schema of the ground term of the nickel ion in $NiCr_2O_4$ was calculated. The parameters of the interaction with the electric field were determined, and the distribution pattern of the electric dipole moments over different positions of nickel in the unit cell was calculated. The model of the $NiCr_2O_4$ magnetoelectric structure at $T < T_c$ was constructed taking into account the data on neutron scattering and the results of the electric polarization measurements. The origin of the magnetodielectric effect was attributed to the peculiarities of the ground state of the nickel ion.

**Keywords:** magnetodielectric effect; spin–orbit coupling; crystal field

## 1. Introduction

Nickel chromite $NiCr_2O_4$ belongs to a large group of compounds that, at high temperatures, have the same crystal structure as the mineral spinel $MgAl_2O_4$. The history of the discovery of these minerals and the many important physical properties of these compounds were described in a recent overview [1]. However, despite the similarity of the crystal structure at high temperatures, chromite nickel has unique physical characteristics and its own applications. The melting temperature is approximately 2300 K, which enables these materials to be used at very high temperatures. $NiCr_2O_4$ chromite exhibits giant magnetostriction [2]. $NiCr_2O_4$ displays the so-called "jumping effect", which is the result of atomic displacements occurring during the phase transition [3]. Nickel chromite compounds are well known as catalytic materials [4]. Investigations in this direction are continuing [5,6].

The expansion of its field of application has stimulated scientific research on nickel chromite. Among the most-important results obtained in recent years, we note the discovery of a large magnetoelectric coupling in nickel chromite [7] and the magnetodielectric effect [8], the microscopic nature of which has not yet been elucidated. The purpose of this paper was to contribute to the microscopic theory of these phenomena, an understanding of which is expected to be important for further applications of nickel chromites in quantum electronics and spintronics.

For a more concrete indication of our tasks, we recall the relevant information about the microscopic structure of nickel chromite, which has been accumulated since the middle of the last century [9,10].

$Cr^{3+}$ ions are in an octahedral position (B) and are not active with respect to the applied electric field. The ground state is orbitally nondegenerate and is characterized by the spin $S = 3/2$. $Ni^{2+}$ ions are tetrahedrally coordinated (A) by oxygen. The ground state in the cubic crystal phase is the orbital triplet ($^3\Gamma_4$). The structural phase transition from the cubic to the tetragonal phase occurs at $T = 320$ K. This transition is commonly interpreted to be caused by the Jahn–Teller effect. The most-pronounced local distortions occur near nickel ions. The tetrahedral fragments of $NiO_4$ are elongated along the **c**-axis of the crystal [3,11,12]. How the $Ni^{2+}$ ($^3\Gamma_4$) state splits is not yet known. This is the first task that will be considered in this paper.

At $T_{FIM} = 74$ K (ferrimagnetic order), the magnetic moments of $Cr^{3+}$ and $Ni^{2+}$ are ordered. The total magnetic moment per the $NiCr_2O_4$ formula unit is much less than 2 $\mu_B$, indicating a ferrimagnetic noncollinear ordering of the magnetic moments. At $T_{AF} = 32$ K, the derivative of the magnetization curve changes abruptly, which has been interpreted as a phase transition to a new phase, which is considered collinear and qualifies as antiferromagnetic. The magnetic moment then increases again and reaches about 0.2 $\mu_B$ at $T = 4.2$ K [13]. Determining the details of the magnetic structure of $NiCr_2O_4$ has turned out to be a difficult task. The first studies of its structure by neutron diffraction methods were carried out in 1972 [14] and have continued in the past decade [13,15]. The chromium ions, which form the main exchange-coupled group, have strong frustration in the spin direction, which makes it very difficult to establish their equilibrium magnetic structure. Recent studies to clarify the crystal lattice structure [16] have been stimulated by the discovery of anomalously large electric polarization in the magnetically ordered phase of $NiCr_2O_4$ [7], the origin of which remains unclear. This is the second problem, which we discuss in this paper.

The third part of our work aimed to clarify the role of the geometric frustration in the crystalline sublattice of nickel ions. In the orthorhombic space group, the position *Fddd Ni 8b* ions are divided into two groups. The group of four equivalent ions containing position $(1/8, 1/8, 1/8)$ we denote as Ni1, and those containing position $(7/8, 7/8, 7/8)$ we denote as Ni2. From the physical point of view, the difference between positions Ni1 and Ni2 can be explained as follows. In a unit cell, there are two $NiO_4$ fragments, which are rotated around the **c**-axis of the crystal by $90°$. Below, we investigate how this affects the distribution of electric dipole moments in $NiCr_2O_4$. As a result, we propose a model of the spatial distribution of the electric dipole moments in a unit cell, which allowed us to interpret both the information about the magnetic structure and the results of the electrical polarization studies within a single model, i.e., we propose a possible model of the magnetic and electrical structure of $NiCr_2O_4$ at $T < 74$ K.

Finally, we describe a microscopic model of the origin of the magnetodielectric effect in $NiCr_2O_4$ and propose the optimal direction of the external magnetic field for the amplification of this effect.

## 2. The Crystalline Field Parameters and Energy Level Schema of the $Ni^{2+}$ Ions

Taking into account the orthorhombic symmetry of the nickel ion positions [16] at $T < 32$ K, we started the calculation of the energy level schema of the $^3F$ ground term by diagonalizing the energy operator:

$$H_{eff} = B_0^{(2)} C_0^{(2)} + B_2^{(2)} (C_2^{(2)} + C_{-2}^{(2)}) + B_0^{(4)} C_0^{(4)} + B_4^{(4)} (C_4^{(4)} + C_{-4}^{(4)})$$
$$+ B_2^{(4)} (C_2^{(4)} + C_{-2}^{(4)}) + \lambda \mathbf{LS} + I_c S_z + I_a S_x. \tag{1}$$

Here, $C_q^{(k)}$ are the crystal field operators, and they are related to the spherical functions by the expression:

$$C_q^{(k)} = \sqrt{\frac{4\pi}{2k+1}} \sum_i Y_{k,q}(\theta_i, \phi_i). \tag{2}$$

The summation is carried out over all electrons of the $Ni^{2+}$ outer shell $(3d^8)$. In the superposition model, when the energy of the system is defined as the sum of the energies of individual pairs, the crystal field parameters are defined by the expression:

$$B_q^{(k)} = \sum a^{(k)}(R_j)(-1)^q C_{-q}^{(k)}(\theta_j, \phi_j), \tag{3}$$

where $j$ refers to the surrounding lattice ions and $a^{(k)}(R_j)$ represents the so-called intrinsic parameters from the lattice ions (ligands) separated from the magnetic ion by distance $R_j$. They were calculated in the same approximation as in [17] for the $Fe^{2+}$ ion in $FeCr_2O_4$. The

summation for the surrounding ions was performed using the structural data reported in [16] at $T = 10$ K. The **x**, **y**, and **z** coordinate system axes were chosen along the crystallographic axes **a**, **b**, and **c**, respectively. As a result of the calculation, the following values (in cm$^{-1}$) were obtained for the Ni1 position:

$$B_0^{(4)} = -6000,$$
$$B_4^{(4)} = 0.458\, B_0^{(4)},$$
$$B_2^{(4)} = 0.019\, B_0^{(4)},$$
$$B_0^{(2)} = -600,$$
$$B_2^{(2)} = -280.$$

The difference between the coordinations of the Ni1 and Ni2 positions is explained in Figure 1. The imaginary parts of the crystal field parameters $B_2^{(4)}$ and $B_2^{(2)}$ are zero. The results of the diagonalization of the operator (1) on the basis of the states of the ground term $^3F$ are reported in Table 1. The energy values in the first column correspond to the cubic phase and those in the second column to the tetragonal phase. The fourth column displays the energy levels of both the Ni1 and Ni2 ions in the magnetically ordered phase, taking into account the orthorhombic distortions. The third, for comparison, illustrates the situation for the tetragonal phase. According to [18], the critical temperature of the orthorhombic distortions is lower than $T_{FIM}$. The last line of Table 1 displays the calculated values of the g-tensor components (in other words, the magnetic moment of the ground state in units of $\mu_B$; see Section 3) and the electric dipole moments in $|e|A$ (see Section 4).

**Table 1.** The calculated energy levels with $B_0^{(4)} = -6000$ and $\lambda = -324$ (in cm$^{-1}$). Other parameters were taken as follows. (a) The approximation of the ideal tetrahedron in the paramagnetic state with $B_{0,\pm2}^{(2,4)} = 0$, $B_{\pm4}^{(4)} = \sqrt{5/14}\, B_0^{(4)}$, and **I** $= 0$. (b) $B_0^{(2)} = -600$, and $B_{\pm4}^{(4)} = -2751$. (c) The tetragonal phase in the magnetically ordered state $I_a = 180$ and $I_c = 3$. (d) The orthorhombic phase in the magnetically ordered state $B_{\pm2}^{(2)} = -280$ and $B_{\pm2}^{(4)} = -115$.

| (a) | (b) | (c) | (d) |
|---|---|---|---|
| 5087 | 5954 | 6160 | 6155 |
| 5087 | 5954 | 6010 | 6010 |
| 4584 | 5915 | 5820 | 5816 |
| 4584 | 3778 | 3896 | 3892 |
| 3748 | 3413 | 3490 | 3495 |
| 3748 | 3413 | 3445 | 3441 |
| 2848 | 3379 | 3405 | 3395 |
| 2300 | 3157 | 3259 | 3271 |
| 2266 | 2926 | 3058 | 3040 |
| 2266 | 2911 | 3044 | 3037 |
| 2172 | 2911 | 2841 | 2854 |
| 2172 | 2882 | 2829 | 2816 |
| 1441 | 1344 | 1435 | 1442 |
| 1441 | 1185 | 1285 | 1255 |
| 1270 | 1185 | 1243 | 1241 |
| 1270 | 1014 | 1041 | 1048 |
| 433 | 725 | 751 | 755 |
| 433 | 338 | 434 | 449 |
| 175 | 338 | 428 | 415 |
| 175 | 299 | 289 | 251 |
| 0 | 0 | 0 | 0 |
| $g_a = 0$ | $g_a = 0$ | $g_a = -2.179$ | $g_a = -1.902$ |
| $g_c = 0$ | $g_c = 0$ | $g_c = -0.037$ | $g_c = -0.040$ |
| $d_b = 0$ | $d_b = 0$ | $d_b = 5.0 \times 10^{-4}\ |e|A$ | $d_b = 5.5 \times 10^{-4}\ |e|A$ |

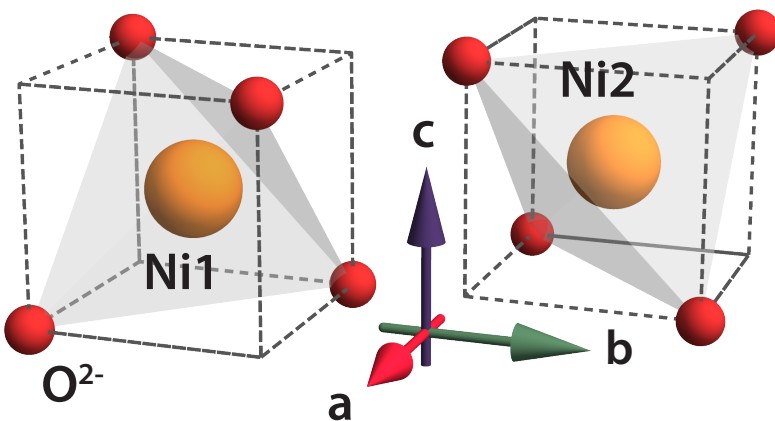

**Figure 1.** The fragments of the $NiO_4$ structure, which demonstrate two possible sites of the Ni ions in $NiCr_2O_4$.

The formation of the energy scheme of the lower levels can be explained as follows. In the absence of distortions of the tetrahedral coordination of the nickel ions (and $\lambda = 0$), the $^3F$ term splits into two orbital triplets ($^3\Gamma_4$, $^3\Gamma_5$) and an orbital singlet, $^3\Gamma_2$. The orbital triplet $^3\Gamma_4$ has the lowest energy. When the spin–orbit interaction is taken into account, the orbital triplet $^3\Gamma_4$ splits into three multiplets with the effective moments $J = 2$, $J = 1$, and $J = 0$. Since $\lambda < 0$, the lower multiplet has $J = 2$. This multiplet is then split under the influence of the orthorhombic components of the crystal field and the exchange field, acting from the surrounding magnetic ions.

The operator $\lambda \mathbf{LS}$ entering Equation (1) stands for the spin–orbit coupling inside the $Ni^{2+}$ electronic shell and $I_\alpha$ are the molecular field parameters of the superexchange interaction between the Ni and Cr pairs. According to the neutron scattering data [16], there is no $I_b$ component at $Ni^{2+}$ sites

Note that the influence of the orthorhombic distortions on both the values of the g-factors and the electric dipole moment is quite weak. We recall that the electric polarization appears already at $T \sim 100$ K [7], i.e., in a magnetically ordered phase. The wave function of the ground state in the quantization scheme $|M_L \, M_S\rangle$ has the form:

$$|\psi_0\rangle = C_1\Big(|3,1\rangle + |-3,1\rangle\Big) + C_2\Big(|-1,1\rangle + |1,-1\rangle\Big) + C_3|0,0\rangle, \tag{4}$$

where the coefficients are $C_1 = 0.492$, $C_2 = 0.323$, and $C_3 = 0.554$. The values of these coefficients under the possible corrections of the parameters can change, but the wave function construction is such that the matrix elements $\langle\psi_0|L_\alpha + 2S_\alpha|\psi_0\rangle = 0$ for all $\alpha = x, y, z$. That is, it is a nonmagnetic singlet. This result fundamentally changes the interpretation of the magnetic moment of the nickel in $NiCr_2O_4$ accepted in the literature. In previous studies, the spin–orbit interaction of the nickel 3d-electrons was ignored, and it was assumed that the ground state had the spin $S = 1$.

## 3. The Effective Magnetic Moment of Nickel

When magnetic moment correlations appear and in the magnetically ordered phase, the exchange interaction with the surrounding spins (molecular field) is added to the energy operator (1). The ground state of the nickel acquires a magnetic moment. When the value of the exchange field parameter is of the order of the Neel temperature, $T_N = 74$ K, the induced magnetic moment is comparable to that obtained from the analysis of the experimental data on the temperature dependence of the magnetization and the data on the neutron scattering (see Table 2).

**Table 2.** The magnetic moment of Ni according to the neutron scattering data (in $\mu_B$).

| Ref. | Bertaut et al. (1972) [14] | Tomiyasu et al. (2004) [18] | Reehuis et al. (2015) [15] | Stüsser et al. (2018) [13] |
|---|---|---|---|---|
| $\mu_a$ (in $\mu_B$) | 2.0, $T < T_N$ | 3, $T < T_N$ | 1.83, $T = 2$ K <br> 1.69, $T = 36$ K | 2.2(1), $T = 1.6$ K |

The origin of the induced magnetic moment on the $Ni^{2+}$ ion can be interpreted as follows. It is related to the virtual excitations resulting from a combination of the action of the exchange and external magnetic fields. For values of the exchange field parameter $I_\alpha$ lower than the excitation energy, one has the following expression for the induced magnetic moment (in $\mu_B$):

$$\mu_\alpha = -\sum_{n\neq0} \langle\psi_0|I_\alpha S_\alpha|\psi_n\rangle\langle\psi_n|L_\alpha + 2S_\alpha|\psi_0\rangle/(E_n - E_0) + H.C. \tag{5}$$

This expression reproduces well the result of the numerical calculation via diagonalizing the energy matrix on the basis of all the states of the $Ni^{2+}$ ($^3F$) term. Figure 2 shows that the nickel magnetic moment is approximately proportional to the exchange field parameter and can reach values typical of a single spin. Note that our scenario for the origin of the $Ni^{2+}$ magnetic moment explains the reason for the differences in the magnetic moment values in the ferrimagnetic (at 36 K) and antiferromagnetic (at 2 K) phases obtained in [15] (see Table 2). In addition, an interesting feature of Figure 2 is that, in the external magnetic field along the **a**-axis, a notable quadratic effect over the applied magnetic field is expected. This effect could be tested experimentally.

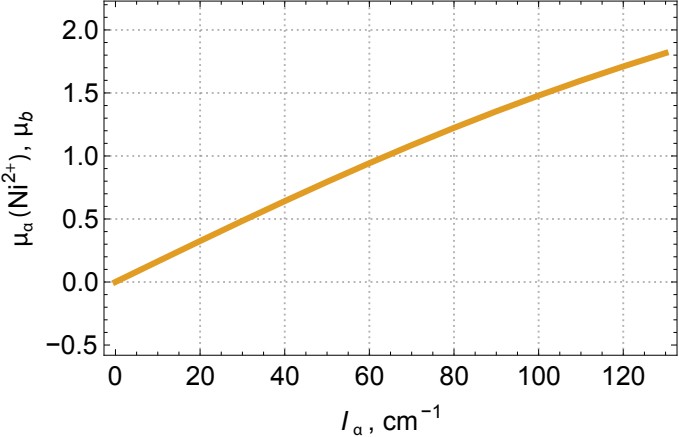

**Figure 2.** The calculated $Ni^{2+}$ magnetic moment as a function of the molecular exchange field **I**.

The reason for the phase transition at $T = 32$ K can be explained as follows. As the temperature decreases, the effective exchange field of the nickel ions from the chromium ions increases. In the first approximation, we can assume that the temperature dependence of the exchange field is proportional to the magnetization. This, following Expression (4), leads to an increase in the magnetic moment of the nickel and, consequently, to an increase in the exchange coupling of the nickel and chromium ions. As a result, the frustration in the orientation of the chromium spins is suppressed, and the noncollinear phase of the chromium magnetic moments is replaced by another phase, which is more stable due to the enhanced antiferromagnetic coupling of the magnetic moments of the chromium and nickel.

## 4. The Relationship between the Electric Dipole Moments of Ni1 and Ni2

The absence of inversion symmetry at the nickel positions leads to the non-zero odd crystal fields, which mix the states of the $3d^8$ ground configuration with the excited configurations of opposite parity $3d^54p$, as well as with the states with charge transfer from

the oxygen ions to the nickel ions. The interaction of the 3d-electrons with the electric field at both positions A and B can be described by the following effective Hamiltonian:

$$H_E = \sum_{p,t,k} \left\{ E^{(1)} U^{(k)} \right\}_t^{(p)} D_t^{(1k)p}. \tag{6}$$

Here, the curly brackets denote the direct (Kronecker) product of the spherical components of the electric field $E_0^{(1)} = E_z$, $E_{\pm 1}^{(1)} = \mp (E_x \pm i E_y)/\sqrt{2}$, and the unit tensor operator $U_q^{(k)}$. The summation indices take the values $k = 0, 2, 4$, $p = 1, 3, 5$, and $t = 0, \pm 3$. In the superposition model, the following relation holds:

$$D_t^{(1k)p} = \sum_j d^{(1k)p}(R_j)(-1)^t C_{-t}^{(p)}(\theta_j \, \psi_j). \tag{7}$$

As a result of the calculation, similar to that performed for $FeV_2O_4$ and $FeCr_2O_4$ in [19] and [17], we obtained (in $|e|A$)

$$d^{(12)3} = -0.590, \qquad d^{(14)3} = 0.824. \tag{8}$$

The role of the contributions from the distant environment in Expression (7) is rather weak; therefore, we neglected it. Note that, for calculation of the values $d^{(12)3}$ and $d^{(14)3}$, we took the dielectric permittivity $\epsilon' = 7.85$ from [20]. Performing the summation over the nearest-neighbor ions, we found that only the imaginary parts of the structural factors were non-zero:

$$Im \sum_{j=1}^{4} C_2^{(3)}(\theta_j \, \phi_j) = \mp 2.0853. \tag{9}$$

It was positive for Ni2, but negative for Ni1. This means that, after averaging the operator (6) on the wave functions of the nickel ions, we obtained electric dipole moments on the Ni1 and Ni2 sublattices of opposite directions. Note that the conclusion about the opposite directions of the dipole moments on the Ni1 and Ni2 sublattices remains valid for the tetragonal phase as well. In the case of the undistorted tetrahedral environment (cubic phase at $T > 320$ K), the structural factor (9) is zero.

## 5. The Relationship between the Electric and Magnetic Moment Components of Nickel

Let us first explain the choice of the exchange field parameters acting on the nickel ions from the side of the chromium and nickel ions, i.e., the values $I_a$ and $I_c$, which we used in our calculations. The choice of $I_a$ is obvious, since the estimated magnetic moment of $Ni^{2+}$ is proportional to it (see Figure 2). In [18], the analysis of the neutron scattering led to the conclusion that, along with the large longitudinal component, it also has a small transverse component (about 0.1 $\mu_B$) along the **c**-axis of the crystal. A recent paper [13] reported the presence of only $\mu_a$ component on chromium ions. The presence of the $\mu_c$ component on the nickel ions was not indicated. However, in the absence of this component, it is difficult to explain the electric polarization induced by the magnetic ordering in $NiCr_2O_4$ [7]. By diagonalizing the sum of the energy operators (1) and (6), we found that, at $I_c = 0$, the nickel electric dipole moment was absent. To overcome this contradiction, we assumed the presence of the $\mu_c$ component on the nickel ions within the error bar for the magnetic moment measurements in [13]. As can be seen from Table 2, the error bar was about 0.1. Therefore, below, we considered values of $I_c$ for which the $g_c$ of the nickel ion is about 0.1 $\mu_B$.

The relationship between the magnetic moment and the electric dipole moment components calculated by diagonalizing the sum of the energy operators (1) and (6) is illustrated in Table 3.

**Table 3.** The calculated values of $g_a$, $g_c$, and $d_b$ in the magnetically ordered state.

| $I_a$ | 60 | 80 | 100 | 120 | 140 | 160 | 180 |
|---|---|---|---|---|---|---|---|
| $-g_a$ | 0.921 | 1.198 | 1.455 | 1.689 | 1.901 | 2.092 | 2.262 |
| $-g_c$ | 0.065 | 0.0628 | 0.060 | 0.058 | 0.055 | 0.052 | 0.049 |
| $10^2 d_b$ ($\|e\|A$) | $-0.0204$ | $-0.0262$ | $-0.0312$ | $-0.0354$ | $-0.0389$ | $-0.042$ | $-0.044$ |

Roughly speaking, the anisotropy of the magnetoelectric effect, i.e., the components of the electric dipole moment, are proportional to the corresponding component of the exchange field acting on the nickel ion. Since the dipole moments on the Ni1 and Ni2 nickel ions are oriented in opposite directions in the absence of an external electric field, the electric polarization $P = 0$. In [7], the presence of the dipole moments in $NiCr_2O_4$ was discovered as follows. At a low temperature, an external electric field was applied and then removed after some time, and the sample was heated uniformly (5 K/min) while measuring the $P$ value. At starting strengths of the electric field $E_{poling}$ = 300 kV/m and $E_{poling}$ = 600 kV/m, in the temperature range of 8–40 K, the $P$ value was almost constant at 13 μC/m$^2$ and 33 μC/m$^2$, respectively. With a further increase in temperature, $P$ decreased and reached zero at $T \sim 110$ K. According to our estimations, for the case of the parallel dipoles on the Ni1 and Ni2 positions, $P$ can reach the value of $\sim$90 μC /m$^2$. Note that the mechanism we considered can yield an even greater value of polarization, as in the case of $Fe_2Mo_3O_8$ [21], for example. In the present case, it is weakened due to the peculiarities of the crystal structure. The expressions (8) show that the internal parameters of the interaction with the electric field have different signs. This feature leads to the reduction in the dipole moments on the nickel ions.

It can be seen from Table 3 that the direction of the electric dipole moment is orthogonal to the plane in which the exchange field lies ($I_a$ and $I_c$). The calculated values of the electric dipole moment are approximated by the expression:

$$d_b = d_\eta \, g_a g_c + d_\varepsilon \, (g_a^2 - g_c^2), \tag{10}$$

where (in $|e|A$) $d_\eta \simeq -3.2 \times 10^{-3}$ and $d_\varepsilon \simeq -2.8 \times 10^{-5}$.

The relation (10) resembles the expression for the electric dipole moment induced by the spin ordering of iron in $FeCr_2O_4$ from [17]. The difference is that, in the present case, the magnetic moment is proportional to the corresponding components of the exchange field parameters (see Table 3).

To summarize our findings, we present a picture displaying the interrelation of the magnetic moments and the electric dipole moments of the nickel ions within a unit cell (Figure 3).

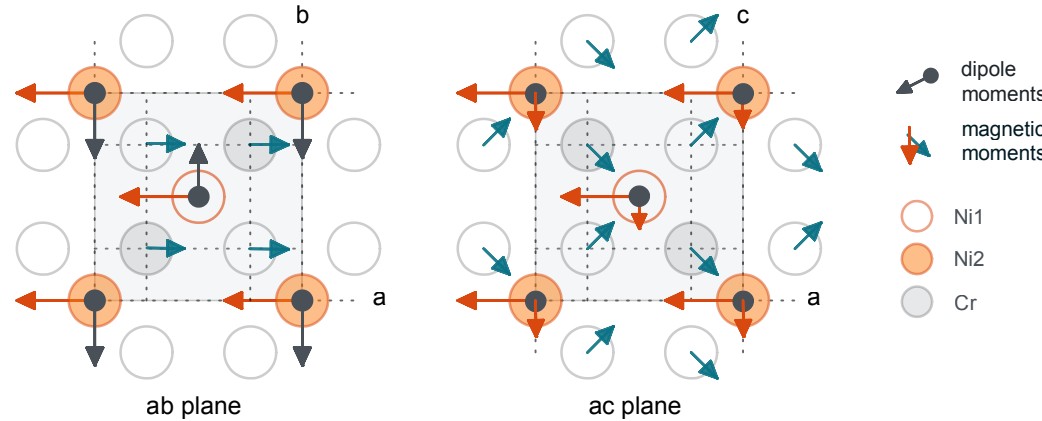

**Figure 3.** The schematic projections of the magnetoelectric structure onto the *ab*- and *ac*-plane (the orthorhombic distortions of the lattice are not visible). It was assumed that the magnetic moments of Ni1 and Ni2 are directed along the **a**-axis (as shown in Figure 10 in [15]). The coordinates of the ions in the unit cell are given according to [16].

## 6. The Origin of the Magnetodielectric Effect

The dependence of the dielectric permittivity on the applied magnetic field has been observed in several works [8,20,22,23]. This interesting effect can be interpreted as follows. When a magnetic field is applied, the exchange field on the nickel ions increases. As a consequence, the electric dipole moment increases. To calculate the magnitude of this effect, we supplemented the sum of the energy operators (1) and (6) by the operator of interaction with the external magnetic field $\mu_B(\mathbf{L} + 2\mathbf{S})\mathbf{B}$ (1).

Figure 4, as well as Table 4 show the results of our calculation for two variants of the magnetic field direction: along the **a**-axis and the **c**-axis. It can be seen that, as the magnetic field increases, the electric dipole moment of the nickel increases monotonically; hence, the dielectric permittivity increases. Our calculations showed that the change in dielectric permittivity for a polycrystalline sample with an increase in the magnetic field by 1 Tesla is of the order of the few present, which is of the order of magnitude consistent with the experimental data [8]. It is interesting to note that, according to our calculations, the magnetoelectric effect was expected to be strongest when the magnetic field is applied along the **c**-axis of the crystal (see Figure 4). It would be interesting to check this conclusion experimentally.

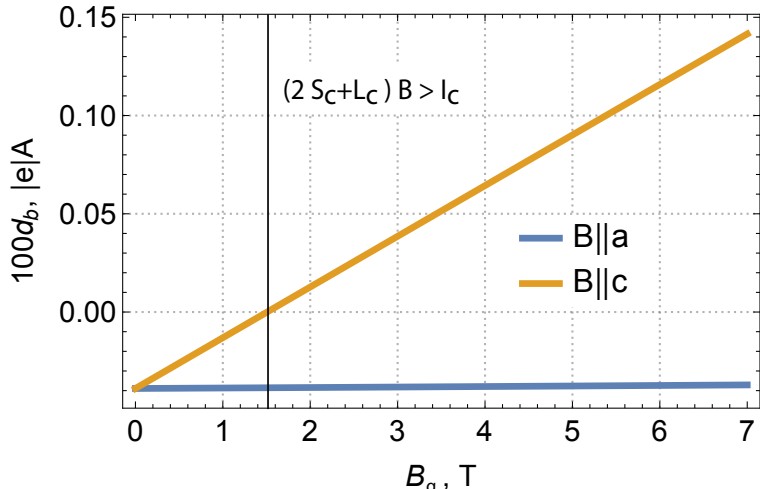

**Figure 4.** The calculated field dependency of the induced dipole moment at the Ni1 site along the **b**-axis. We assumed that the crystal was magnetized along the **a**-axis (a) with $I_a = 180$ cm$^{-1}$ and $I_c = 3$ cm$^{-1}$.

**Table 4.** The calculated field dependency of $1000 \cdot d_b$ ($|e|A$) in the magnetically ordered state. We assumed that the crystal was magnetized along the **a**-axis (a) with $I_a = 180$ cm$^{-1}$ and $I_c = 3$ cm$^{-1}$.

| B, T | 0 | 0.5 | 1 | 1.5 | 2 | 2.5 | 3 |
|---|---|---|---|---|---|---|---|
| B ‖ a | −0.065 | −0.065 | −0.064 | −0.064 | −0.064 | −0.064 | −0.064 |
| B ‖ c | −0.065 | −0.052 | −0.039 | −0.026 | −0.013 | −0.0 | 0.013 |

## 7. Concluding Remarks

The main results of our work can be summarized as follows. The crystal field parameters and the energy level schema of the ground term of the nickel ions were calculated. It was found that the ground state of nickel is nonmagnetic when the spin–orbit interaction is taken into account. In magnetically ordered phases, the magnetic moment is induced by the exchange field and turns out to be sufficiently large. For the exchange field parameters on the order of the Curie temperature, the magnitude of the calculated magnetic moment corresponds to the value of $\mu_a$ determined from the neutron scattering. The distribution of the electric dipole moments at different positions of nickel in the unit cell was calculated. It was found that the electric dipole moment $d_b$ on the nickel ions arises only in the presence of

an additional magnetic moment component $\mu_c$. It was shown that it is sufficient to assume the presence of a longitudinal component of the magnetic moment $\mu_c \sim 0.1\ \mu_B$ to match the neutron scattering data and the results of the electric polarization measurements. As a result, a variant of the magnetoelectric structure of the $NiCr_2O_4$ at $T < T_c$ was suggested, which allows one to explain both the data of the neutron scattering study and the results of the electric polarization measurements.

In addition, we were able to establish a microscopic model of the origin of the magnetodielectric effect and specify the direction of the magnetic field to enhance this effect.

**Author Contributions:** Conceptualization, writing—original draft preparation, investigation, M.E.; visualization, investigation, K.V. All authors have read and agreed to the published version of the manuscript.

**Funding:** This work was supported by the Russian Science Foundation (Project No. 19-12-00244).

**Data Availability Statement:** Not applicable.

**Conflicts of Interest:** The authors declare no conflict of interest.

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
