# Peer review of "The Origin of the Magnetic and Electric Dipole Moments of Ni2+ in NiCr2O4"

_condensedmatter, doi:10.3390/condmat8010023_

Round 1

Reviewer 1 Report

MDPI condensed matter 2188571

Interrelation of Magnetic and Electrical Structures in NiCr2O4

By Mikhail Eremin and Kirill Vasin

In their theoretical investigation Eremin and Vasin calculate the electronic level scheme of Ni2+ in the distorted spinel compound NiCr2O4 including spin-orbit coupling and the exchange fields imposed on Ni2+ by the Cr3+ magnetic moments. Using their wave functions the authors calculate by perturbation theory the magnetic moment. In addition, they evaluate the electric dipole moment of Ni2+ in the non-centrosymmetric environment and make a conclusion about the orientation of the Cr3+ imposed exchange fields.

The essential finding of their investigation is that by including spin-orbit coupling, Ni2+ in a tetrahedral coordination (regular tetrahedron, tetragonal or orthorhombic distortion) the ground state is a nonmagnetic singlet. The magnetic moment is induced by the Cr3+ exchange field and/or an external magnetic field, and it results from virtual excitations to excited states. The calculated magnetic moments agree quite well with recent neutron scattering results. In order to justify the presence of an electric dipole moment the presence of an I_z component of the exchange fields is proposed.

The results reported in the manuscript are able to shed a completely new light on the properties of Ni2+ in the distorted spinel compound NiCr2O4 and may substantially help to understand the complex magnetic, magnetostructural and magnetodielectric behavior of NiCr2O4. The manuscript merits acceptance after revisions listed below:

The revisions concern less the physical methods or the results but more the fairly superficial way of manuscript editing. In some cases it is just due to simple typos, in other cases it may lead to confusion or even to contradictions:

1.    For example, page 2 and 3 lines 66 to 68: The text says that the third column in Table 1 ‘illustrates the situation for the tetragonal phase with “short range correlations”. The last line displays the calculated values of the g-tensor components and the electric dipole moments.’

As a matter of fact, the column (c) in Table 1 shows the energies with exchange fields included. What is the reason that the authors imply ‘short range correlations’? The last line in Table 1 does not report g-tensor components or electric dipole moments. g-tensor components are given in the caption to Table 1 where they are a bit misplaced because they are calculated later in chapter 3 of the manuscript.

2.    The authors should provide a source for the exchange field components I_x and I_z. Where do these values originate from?

3.    Page 2 line 58: ‘the following values … were obtained for the Ni1 position’. I understand from Ref. 6 (Suchomel, et al.) that in the orthorhombic phase Ni occupies only one position (8a in space group Fddd). For a less familiar reader this can lead to confusion.

4.    At many places in the manuscript the authors use the term ‘rhombic’ distortion. I understand that the low-temperature phase is orthorhombic and the distortion is rather ‘orthorhombic’

5.    Page 3, line 74 the authors say that the ‘lower multiplet has J=2’ This is not correct. Following the text  it is rather a singlet, as one can also see from the energy levels listed in Table 1.

6.    Caption to Table 1: a reference should be given for the spin-orbit coupling parameter. Abragam-Bleaney, citing Blume/Watson quote a value of -343 cm-1. Why do the authors chose -300 cm-1?

Finally, as found in Ref. (3) in the orthorhombic phase the Cr atoms order with two disparate magnetic phase, one of them is incommensurate. A discussion whether this could also be the reason for the presence of I_z and the electric dipole moment would be instructive.

Additional comments:

In a number of cases definite and indefinite articles and missed words have to be included and some typos have to be corrected:

p.5, line 110: add ’electric’ before ‘dipole moment’

p. 7, line 152: add ‘is’ before ‘proportional’

p. 7, line 156f: add ‘field’ after ‘magnetic’

p. 6, line 129+130: replace ‘bare’ by ‘bar’

Suggestions:

In my opinion, using ‘Electronic’ instead of ‘Electric’ in the title characterizes the contents of the manuscript much better.

In summary, the manuscript should be accepted after a revision along the points listed above.

Author Response

We are grateful for your remarks and suggestions. Here is the list of changes we made as well as our comments

1. We  have dropped   phrase “short range correlations” . Component locations corrected. The components of g-tensor were moved, the description was changed.

2. Explanation of the molecular exchange field was added after Eq. (1)

3. Here we would disagree. The symbol Ni8a means that there are 8 nickel ions in a unit cell at A-site position.  8b positions are divided into two groups.   The group of four ions containing position 9 we denote as 1 , and the group with position 2 as . These 8 positions are formed two groups  by 4 ions on the basis of the two points at (1/8, 1/8 1/8) and (7/8,7/8,7/8). For short, we refer to the nickel ions in these two groups as Ni1 and Ni2 .  But to clarify it we have added this explanation in to the  introduction.

4. This is right. We replaced ‘rhombic’ with ‘orthorhombic’. 

5. We would not agree with the statement of the referee. The spin-orbit coupling parameter has a negative sign, therefore the lowest energy multiplet has J=2.  Anyway thanks for this comment, it forced us to check and refine the energy levels table.

6. Yes, Abragam-Bleaney, citing Blume/Watson quote of a value of -343 cm-1,   but it is for free ions. In the same table authors gave also an experimental value -324 cm-1. We have chosen the last one. The one, which was written in the text before (-300cm-1) was a typo. Now it is correct.

Finally, as found in Ref. (3) in the orthorhombic phase the Cr atoms order with two disparate magnetic phase, one of them is incommensurate. A discussion whether this could also be the reason for the presence of I_z and the electric dipole moment would be instructive. 

It is too early to make any conclusions about the origin of the exchange field component along the crystal с- axis.  The accuracy of measurements of this component in neutron scattering experiments is still insufficient.  There are problems of growing single-crystal samples of sufficiently large size.   Our result in this respect is that the Iz component is present, otherwise the magnetoelectric effect cannot be understood.

p.5, line 110: add ’electric’ before ‘dipole moment’

  1. 7, line 152: add ‘is’ before ‘proportional’
  2. 7, line 156f: add ‘field’ after ‘magnetic’
  3. 6, line 129+130: replace ‘bare’ by ‘bar’

Corrections have been made to the text

In my opinion, using ‘Electronic’ instead of ‘Electric’ in the title characterizes the contents of the manuscript much better.

Referee 3 made a same suggestion. In this regard, we have changed the title of the article.

Reviewer 2 Report

I would recommend the paper for publication after the authors consider

remarks given below.

line 12.

'..remains topical.' -> '..understanding of its properties remains topical'

Formula (1)

Explain the meaning of Ix, Iz  and why Iy is missing. It follows from part 3 that they refers to exchange parameters. Then, however, labeling of horizontal axis in Figure 1 is wrong.

Crystal field terms with subscripts -2 and -4 appear in (1), but  only crystal field parameters with positive subscripts are given below line 58.

On line 68 authors promise that electric dipole moments will be given in Table 1, but they are not.

There is a confusion in labeling the g-factors. Several times they have subscripts a, b, c, several times the subscripts are x, y, z. I understand that this is connected with two different Ni sites and the orthorhombic crystal structure. This should be, however, properly explained and the labeling unified.

Part 6.

'The origine' -> 'The origin'

'..magnetic was..' -> '.. magnetic field was..'

'.. is on the order..' -> '..is of the order.. '  (two times)

I suggest that the hamiltonian (11) is included already to (1).

Author Response

We are grateful to the Referee for all the remarks and suggestion.

line 12.

'..remains topical.' -> '..understanding of its properties remains topical'

This is a good point. we have taken it into account in the introduction

 Formula (1)

Explain the meaning of Ix, Iz  and why Iy is missing. It follows from part 3 that they refers to exchange parameters. Then, however, labeling of horizontal axis in Figure 1 is wrong.

Yes, Ix, Iz  and Iy  refers to exchange parameters.  The explanation is added after formula (1)

Crystal field terms with subscripts -2 and -4 appear in (1), but  only crystal field parameters with positive subscripts are given below line 58.

We use a coordinate system in which the crystal field parameters with indices -2 and -4 are equal to those with positive indices 2 and 4. This is understandable because our parameters are real . i.e. the imaginary parts are equal to zero (which was mentioned in the text as well).

On line 68 authors promise that electric dipole moments will be given in Table 1, but they are not.

The typo was corrected. We give g-factors and dipole moments to below the middle line of the table

There is a confusion in labeling the g-factors. Several times they have subscripts a, b, c, several times the subscripts are x, y, z. I understand that this is connected with two different Ni sites and the orthorhombic crystal structure. This should be, however, properly explained and the labeling unified.

Following the recommendation of the Referee the dipole moments in the Tables and in the text are indexed as a, b and c.

Part 6.

'The origine' -> 'The origin'

'..magnetic was..' -> '.. magnetic field was..'

'.. is on the order..' -> '..is of the order.. '  (two times)

I suggest that the hamiltonian (11) is included already to (1).

Thank you for the remarks. We made all changes to the text.

Reviewer 3 Report

This manuscript presents a theoretical study of the magnetoelectric structures and properties in NiCr2O4. The authors reported Crystalline field parameters and the energy level schema of the Ni ions first. Then by adding the exchange field and electric field parameters in the Hamiltonian, the authors calculated the magnetoelectric structure of NiCr2O4 in the magnetically ordered phase. A microscopic model was proposed as well to explain the origin of the magnetodielectric effect in this material. However, there are some issues that the authors need to address before the paper can be considered for publication.

1.     The paper lacks sufficient literature review and references in the beginning. The first few sentences in the introduction are quite vague. Not much information is provided (and there is no reference at all!). Then the authors raised three questions. However, why are these questions important and challenging? More background information is needed. And the authors need to have better and clear presentation of the importance of these topics they chose.

2.     The authors used T_{FIM} in line 20 without defining it first. Besides, there are multiple transition temperatures in NiCr2O4. It would be better to distinguish them with different subscript and then use these notations consistently throughout the paper without citing specific temperatures.

3.     The word “electrical structure” in the title is uncommon and very confusing. Do the authors mean the electric dipole moment distribution in NiCr2O4? It is better to rephrase the title to make it clear.

4.     In section 2, these are terms with I_z and I_y in eq. (1). However, the authors did not define and discuss these terms until section 5. Besides, the term of I_x is missing.

5.     In lines 66-68, the authors claimed that there are g-tensor components and electric dipole moments reported in Table 1, but they are not given in the table. Besides, it would be more elucidative to show the energy level schema as a figure instead of a table.

6.     The authors need to provide relevant references in line 81 when they compared their results with previous studies.

7.     What are the positions "A" and "B" in the first paragraph in section 4? Do they refer to the Ni1 and Ni2 sites?

8.     It would be better if the authors can provide more explanation on the strong anisotropic field dependence of the magnetoelectric effect.

Author Response

We are grateful to the Referee for the remarks and suggestions. Following the recommendations we have sent our manuscript to MDPI English proofreading service. Here is the list of changes we made to the text as well as our replies to the given comments

  1. The paper lacks sufficient literature review and references in the beginning. The first few sentences in the introduction are quite vague. Not much information is provided (and there is no reference at all!). Then the authors raised three questions. However, why are these questions important and challenging? More background information is needed. And the authors need to have better and clear presentation of the importance of these topics they chose.

The introduction has been expanded.  We have added a paragraph on the use of chromite nickel and added the required references. 

  1. The authors used T_{FIM} in line 20 without defining it first. Besides, there are multiple transition temperatures in NiCr2O4. It would be better to distinguish them with different subscript and then use these notations consistently throughout the paper without citing specific temperatures.

We have added explanation of the abbreviations.

  1. The word “electrical structure” in the title is uncommon and very confusing. Do the authors mean the electric dipole moment distribution in NiCr2O4? It is better to rephrase the title to make it clear.

Following the recommendation of the referee, we corrected the title of article. 

  1. In section 2, these are terms with I_z and I_y in eq. (1). However, the authors did not define and discuss these terms until section 5. Besides, the term of I_x is missing.

In text we added a foot note, that according to the neutron scattering experiments  I_x=0.

  1. In lines 66-68, the authors claimed that there are g-tensor components and electric dipole moments reported in Table 1, but they are not given in the table. Besides, it would be more elucidative to show the energy level schema as a figure instead of a table.

We tried to replace Table 1 by a figure.  However, we have found that the information about levels splitting due to the spin-orbit interaction, Jahn-Teller distortions and exchange field is really missing. Usually, in papers on infrared and terahertz spectroscopy authors prefer to give tables of energy levels (see for example  Ref. [10.1103/PhysRevB.102.115139]), which makes easy to match the excitations with the calculated levels.

  1. The authors need to provide relevant references in line 81 when they compared their results with previous studies.

We are not aware of calculations of the splits of the ground term of the nickel ion taking into account the spin-orbit interaction, Jahn-Teller and lattice distortion , and exchange fields in NiCr2O4.  Magnetic and electric dipole moments on nickel ions have been calculated for the first time. 

  1. What are the positions "A" and "B" in the first paragraph in section 4? Do they refer to the Ni1 and Ni2 sites?

In the introduction we have explained that in the literature the letters A and B are used to denote the positions of Ni and Cr, respectively.

  1. It would be better if the authors can provide more explanation on the strong anisotropic field dependence of the magnetoelectric effect.

After Table 3 we added the following sentence.  Roughly speaking, the anisotropy of the magnetoelectric effect i.e. the components of the electric dipole moment are proportional to the corresponding component of the exchange field acting on the nickel ion. 

Round 2

Reviewer 1 Report

MDPI condensed matter 2188571v2

Interrelation of Magnetic and Electrical Structures in NiCr2O4,

By Mikhail Eremin and Kirill Vasin

The authors have revised their manuscript which is now (after corrections of two typos, which can be done in the proofs; see below) along my criticism and remarks.

As already stated in my preceding Report: ‘The results able to shed a completely new light on the properties of Ni2+ in the distorted spinel compound NiCr2O4 and may substantially help to understand the complex magnetic, magnetostructural and magnetodielectric behavior of NiCr2O4.’

Therefore, I warmly recommend accepting the manuscript as is.

Typos:

page 6, two lines before eqn. (7): replace ‘tge’ by ‘the’

page 9, line 227 replace ‘of’ by ‘for’

Reviewer 2 Report

The authors reacted properly to the remarks of the referee and I now recommend the paper for publication.

Reviewer 3 Report

The revised manuscript has adequately addressed all my initial questions and is now suited for publication.